# Research on learning achievement classification based on machine learning

**Jianwei Dong**[1,2]☯*, **Ruishuang Sun**[3]☯, **Zhipeng Yan**[4], **Meilun Shi**[4], **Xinyu Bi**[4]

**1** College of Educational Science, Xinjiang Normal University, Urumqi, China, **2** College of Educational Science, Xinjiang Teacher's College, Urumqi, China, **3** School of Software, Xinjiang University, Urumqi, China, **4** School of Computer Science and Technology, Xinjiang University, Urumqi, China

☯ These authors contributed equally to this work.
* 1982086289@qq.com

**Data availability statement:** All relevant data are within the article and its supporting information files.

## Abstract

Academic achievement is an important index to measure the quality of education and students' learning outcomes. Reasonable and accurate prediction of academic achievement can help improve teachers' educational methods. And it also provides corresponding data support for the formulation of education policies. However, traditional methods for classifying academic performance have many problems, such as low accuracy, limited ability to handle nonlinear relationships, and poor handling of data sparsity. Based on this, our study analyzes various characteristics of students, including personal information, academic performance, attendance rate, family background, extracurricular activities and etc. Our work offers a comprehensive view to understand the various factors affecting students' academic performance. In order to improve the accuracy and robustness of student performance classification, we adopted Gaussian Distribution based Data Augmentation technique (GDO), combined with multiple Deep Learning (DL) and Machine Learning (ML) models. We explored the application of different Machine Learning and Deep Learning models in classifying student grades. And different feature combinations and data augmentation techniques were used to evaluate the performance of multiple models in classification tasks. In addition, we also checked the synthetic data's effectiveness with variance homogeneity and P-values, and studied how the oversampling rate affects actual classification results. Research has shown that the RBFN model based on educational habit features performs the best after using GDO data augmentation. The accuracy rate is 94.12%, and the F1 score is 94.46%. These results provide valuable references for the classification of student grades and the development of intervention strategies. New methods and perspectives in the field of educational data analysis are proposed in our study. At the same time, it has also promoted innovation and development in the intelligence of the education system.

## Introduction

In the era of big data, education related data has seen a significant increase in content and quantity. How to realize educational data mining has become an important research topic

**Funding:** The author(s) received no specific funding for this work.

**Competing interests:** The authors have declared that no competing interests exist.

in the current education field [1]. Our text effectively utilizes various educational data and employs methods such as relationship mining [2], prediction [3–5], and clustering [6–8] to accurately evaluate and understand student behavior and performance. As well as our text also helps students flexibly adjust their learning strategies. As is well known, academic achievement is of great significance in the modern education system. It serves as a core indicator for measuring students' educational activities and academic performance. Accurately estimating students' academic performance can improve the quality of student education, explore the laws of educational development, and enrich educational models.

In education policy and research, estimating student grades is crucial for analyzing and improving the education system. By analyzing the various characteristics of students, we can understand what influences their academic achievement. And by analyzing data, we can spot future trends in grades and keep an eye on students' academic achievement with early warnings. Currently, structured data in education is becoming increasingly rich and complex. Research [9] has found that students' achievements are not only directly related to their grades in various subjects, classroom conditions, and learning abilities, but also closely related to potential characteristics such as family background, extracurricular activities, and family education. In addition to this, the concept of academic integrity, i.e. honesty, trust, fairness, respect, responsibility and courage, can be explored in terms of its impact on student achievement [10]. Thus, the classification of student achievements should identify and analyze potential learning difficulties from a more comprehensive perspective. And it should find what affects student performance and give advice on how to improve, improving the quality of education [11] and promoting the comprehensive development of students.

Due to the increasing complexity of educational data, the model of classifying student grades needs to have significant advantages in dealing with multidimensional features, especially when dealing with nonlinear features. In addition, in specific small-scale studies or datasets of specific classes, the amount of data is often limited. The current classification methods for academic achievement can be roughly divided into two categories. One is based on statistical methods and machine learning methods. Another method is based on statistics mainly includes linear regression [12], logistic regression [13] and etc. These methods usually have simple principles, convenient calculations, and can quickly obtain the classification results of students' academic achievements. However, the prediction accuracy of these methods is limited, making it difficult to capture complex nonlinear relationships. These methods also don't fully use the rich information in education data and are not good at finding hidden factors or learning from them. Compared to statistical methods, machine learning based methods perform well in handling nonlinear relationships, multidimensional features, and mining complex patterns in data. The more typical K-means clustering and Long Short-Term Memory (LSTM) networks are used to model and predict student performance based on their reported behaviours and preferences [14]. In addition, machine learning methods include decision trees [15–17], support vector machines [18–20], random forests [21], neural networks [22,23], etc. These methods include They are able to better handle the multidimensional features in student achievement prediction, resulting in higher prediction accuracy. However, in a limited number of small datasets, machine learning methods are prone to overfitting. In addition, these data have relatively high requirements in computational complexity, data volume and training time, and are more dependent on data.

Limitations in data volume and multidimensional data can lead to inaccuracies in learning evaluation. These issues also lead to teachers' lack of understanding of students' current

learning status and may mislead educational decisions, seriously affecting the actual quality of education. Therefore, this article addresses issues such as handling nonlinear relationships, multidimensional features, and limited data volume, we propose an innovative method based on radial basis function network for Gaussian distribution based oversampling (GDO-RBFN). This method can effectively alleviate data imbalance and limited data volume. Firstly, we perform Gaussian distribution based sampling on data with limited data volume, and generate samples using the statistical features of the original data. This approach can provide sufficient data for training subsequent models and weaken the impact of class imbalance in the original data. Secondly, we import the generated new dataset into the RBFN model. This approach can demonstrate advantages in predicting student academic performance through its excellent non-linear fitting and generalization abilities [24]. The effective combination of two methods improves the accuracy of predicting students' academic performance. This combination motivates students to plan reasonable learning goals, enhances learning motivation, and it also benefits teachers to teach, optimize teaching methods, and allocate educational resources reasonably. In summary, the contributions of this article are as follows:

- This article proposes a model called GDO-RBFN, which innovatively combines Gaussian based distribution oversampling and radial basis function networks. It expands the original dataset through Gaussian distribution sampling. It effectively alleviates the problems of limited educational data and imbalanced data categories.
- This article uses Radial Basis Function Network (RBFN) as a nonlinear strong fitting model, which can effectively capture high-dimensional features. It generates more representative sample data. More representative sample data is generated. This significantly improved the predictive performance of the GDO-RBFN model in student academic achievement prediction classification tasks.
- We compared six mainstream performance prediction models on an education dataset with class imbalance. We use multiple metrics such as Accuracy, Precision, and Recall for evaluate. The significant advantages of the GDO-RBFN model in handling limited data and mining feature relationships have been validated by us. Compared with other models, our model has better overall performance. From the perspectives of data volume and features, our model solves the practical problem of inaccurate classification of student grades. Our model provides educators with more accurate predictive analysis tools.

The rest of this article is organized as follows: The second section reviews the development and application of data mining methods and GDO technology in the field of education. The third section first introduces the framework of our proposed method GDO-RBFN, and then focuses on describing the effective data augmentation and RBFN model. The fourth section introduced the experimental setup and evaluation criteria, and comprehensively discussed the experimental results. The fifth section introduces the limitations and future development of this study.

## Review

### Development of EDM technology

Educational Data Mining (EDM) technology applies interdisciplinary theoretical knowledge and practical techniques to solve practical educational problems. Including education, computer science, statistics, and computer science, etc. Among them, the most important form is still to extract valuable information from massive amounts of specific structured educational data. Further it needs to analyze and explore students' learning characteristics,

behaviors, emotions, and other factors, as well as the difficulty of the course and the correlation between courses. Finally, it needs to establish an effective analytical model to predict students' learning ability or academic achievement, and conduct accurate assessment and classification. Around the 1980s, it was widely believed from a psychological and cognitive perspective that factors such as student motivation, planning, and learning styles were associated with student performance. Cortez [25] used multiple classification algorithms to make reasonable predictions on the academic performance of Portuguese high school students. His results showed that the decision tree had a significant effect and accurate predictive performance. Peña-Ayala [26] conducted an in-depth analysis of the application of Data Mining (DM) techniques, such as clustering analysis, classification algorithms, and regression models, in the field of education. Anzer [27] predicted individual course grades and uses linear regression to predict final grades. He predicted the factors most relevant to final grades based on his performance before, during, and after class. Hussain [28] evaluated students' academic achievement using 12 characteristics representing academic and personal qualities. He compared RF, PART, BayesNet and other methods, and found that RF still had a significant advantage in prediction accuracy. Minn [29] proposed a novel model that combined multi task learning and graph neural networks. He significantly enhanced the model's understanding and predictive ability of students' knowledge status by modeling the dependency relationships between knowledge points. Meanwhile, the model can evaluate students' mastery level of different knowledge points. Fan Yang [30] proposed using classification BP-NN for student performance estimation. This model can estimate students' attributes by referring to their prior knowledge and the attributes of other students with similar features. Mahmoud [31] analysed and compared the results of each classifier used, including K-mean, maximum likelihood, support vector machines and neural networks, integrating different types of classification. Olabanjo [32] collected students' past learning records and their psychological abilities. Based on this, the RBFN model is trained to predict student performance, and principal component analysis is also used to evaluate performance.

## Application of sampling technology

Sampling techniques play a crucial role in analyzing data and modeling processes, especially in situations with limited samples. Sampling techniques amplify limited samples, generate new data samples, and expand the size of the dataset. Thus the effectiveness of training and the accuracy of prediction are thus improved. For example, the minority undersampling method k-INOS [33], based on the domain of influence approach. It improves the effectiveness of handling imbalanced data by preventing noise samples from contaminating the oversampling process. In predicting amyloid protein models. OPTIC-SMOTE [34] is an improved SMOTE model based on density clustering. It significantly improves the authenticity and representativeness of synthesized samples by removing noisy samples and fully utilizing boundary sample information. ADPPTC [35] can accurately complete tasks within a specific time frame, ensuring optimality and specified time stability. GAME [36] is a generative adversarial method for minority groups. It adjusts the parameters of the local linear model to approach the majority category by extending the sampling margin of the data generation and adversarial stages. Its goal is to enhance the diversity and representativeness of synthesized samples, avoiding the problem of minority class samples being limited to a finite sample space. CHSMOTE [37] is a convex packet-based SMOTE algorithm. This algorithm involves selecting the boundary majority samples as the initial samples. It is sampling and identifying the sample synthesis region by checking whether the constructed convex packet contains boundary majority samples or not. Its algorithmic principle is to generate more samples with valid

information by using expanding the generation range of synthetic samples. Compared to SMOTE-like sampling, Gaussian Distribution based Oversampling (GDO) [38] is a sample generated based on statistical features of the data. GDO simulates different feature patterns and distributions. At the same time, GDO can effectively reduce the influence of noise samples and enhance the diversity of samples. It also effectively avoids the problem that SMOTE-type methods are too dense. In the analysis of structured data in education, class-specific datasets often face the challenge of insufficient data volume. This issue this limits the ability of the model to provide in-depth understanding and accurate assessment of student behaviors and performance. Limited amount of data may lead to insufficient model training. It fails to capture the connections and deeper meaning of students' underlying behaviors. To cope with this problem, this study introduces a proposed novel data resampling technique called GDO. This technology generates virtual student learning records that simulate different modes of learning and subject understanding. These generated data can effectively extend the original dataset and also support the analysis of student learning trends, personalized education needs. These data can also assess the effectiveness of teaching methods. This approach helps to reduce reliance on a limited sample of actual data and provides a more comprehensive and reliable information base for analyzing education data.

## Materials and methods

### Modelling framework

The framework of the model is shown in Fig 1. This study proposes a classification model for GDO-RBFN students' expected achievement. The aim of this paper is to weaken the effect of

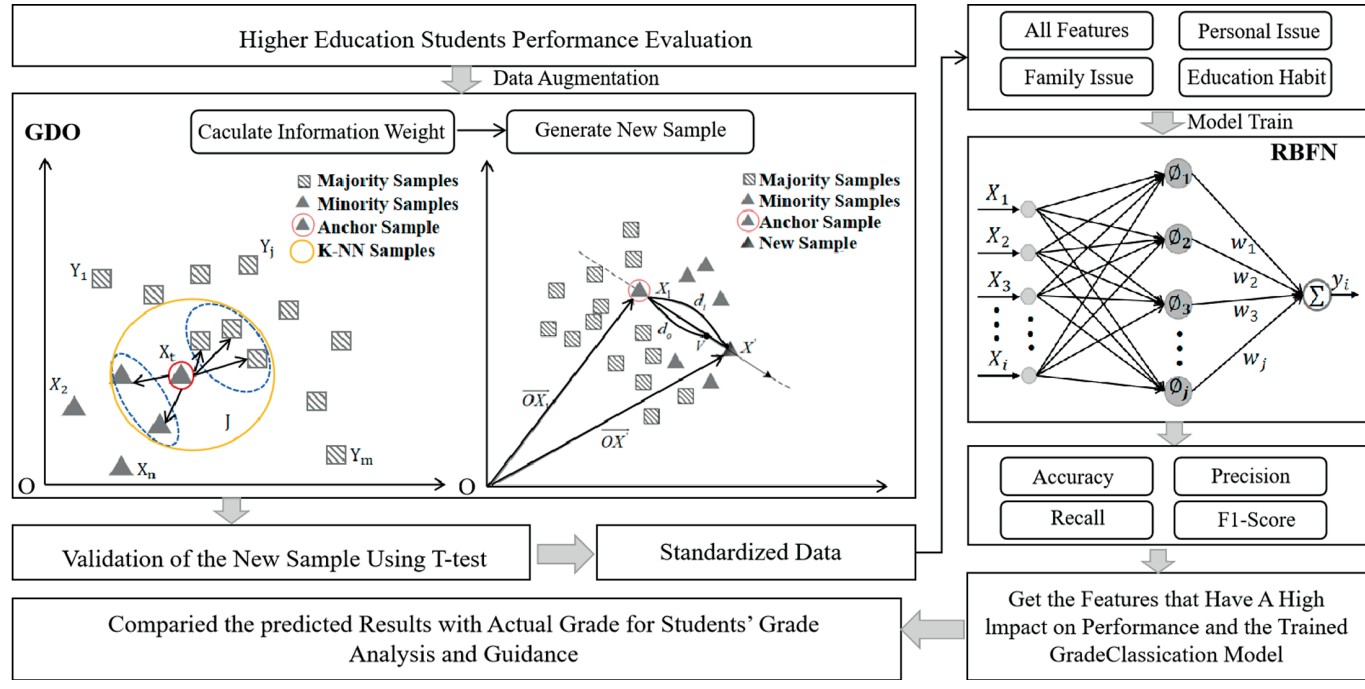

**Fig 1. Classification model for predicting student performance GDO-RBFN.**

class imbalance by exploiting the advantages of RBFN by deep mining the GDO and high-dimensional features that are effectively augmented to the data. This paper is also more effective at capturing the underlying characteristics of the data. The core framework is divided into two parts, the data augmentation GDO part and the RBFN classification part. The GDO method is used to perform data augmentation, calculate information weights and compute new samples. For the generated samples, the validity of the samples was tested using t-tests and the data were standardized. Different characteristics were chosen for the study: four categories: overall characteristics, personal problems, family problems and educational habits. The RBFN model is used for classification and model evaluation to compare predicted scores with actual scores. At the same time, it analyses the results and guides students to performance improvement.

## GDO data amplification

GOD is a diversity sample generated by modelling different feature patterns and distributions based on the statistical characteristics of the data. It is to balance the class distribution of the original data through data-enhanced samples, thus weakening the effect of class imbalance. This GDO is broadly divided into three parts: probabilistic selection of anchor instances, generation of new instances and oversampling probabilistic analysis.

- **Calculate the Anchor Selection Probability for Each Minority Class Instance.**

    Measuring the importance of data information is based on the distance and density of a small number of class instances, and the probability of information weights is calculated as follows:

    Considering that different instances carry different information, we focus on distinguishing the differences between a few instances, set as the density factor $M(X_i)$ and the distance factor $C(X_i)$, included among these, $N_i^{min}$ denotes the set of few class instances in $N_i$. $N_i^{max}$ denotes the set of most class instances in $N_i$. $J^{min} = \{X_1, X_2, \cdots, X_n\}$ belongs to the minority class of real sets. $J^{max} = \{Y_1, Y_2, \cdots, Y_m\}$ belongs to the majority class of real sets. The density factor $M(X_i)$ is defined as follows:

$$M(X_i) = \frac{|N_i^{maj}|}{K}$$

$M(X_i)$ represents the proportion of majority class instances in the K-nearest neighbors of $X_i$. The distance factor $C(X_i)$ is defined as follows:

$$C(X_i) = \frac{\sum_{X_j \in N_i^{maj}} dist(X_i, X_j)}{\sum_{X_j \in N_i^{min}} dist(X_i, X_j) + \sum_{X_j \in N_i^{maj}} dist(X_i, X_j)}$$

$M(X_i)$ and $C(X_i)$ are added together to get the information weight:

$$I(X_i) = M(X_i) + C(X_i)$$

- **Selection of Anchor Instances and New Generation of Samples.**

    Our study uses a roulette wheel algorithm to select anchor instances from a small number of points based on anchor selection probability. We determine the selection probability of each minority class instance as an anchor instance. The selection probability is denoted by $\hat{I}(x)$.

| Minority class | $X_1$ | $X_2$ | $X_3$ | ... | $X_n$ |
|---|---|---|---|---|---|
| Selection probability | $\hat{I}_1$ | $\hat{I}_2$ | $\hat{I}_3$ | ... | $\hat{I}_n$ |

It needs to iteratively pick anchor instances. It generates new samples of minority instances around the anchor instances that follow the Gaussian distribution model until the number reaches the same as the number of majority classes. The quantity to be synthesized is set to $G_a$:

$$G_a = G * |f^{maj}| - |f^{min}|$$

$|f^{min}|$ denotes instances in the minority class. $|f^{maj}|$ denotes instances in the majority class. $G_a$ denotes the sampling rate.

Roulette Algorithm Selection

Our study is to generate a uniformly distributed $\theta$ each time an anchor instance is selected from a small number of instances, and compare it with the cumulative probability. The instance $X_i$ is selected if the following conditions are satisfied:

$$\sum_{i=-1}^{k-1} \hat{I}(X_i) < \theta < \sum_{i=1}^{k} \hat{I}(x_i)$$

In which, each of the few instances $X_i$ is selected as an anchor instance the number of times $W(X_i)$ expects:

$$E(W(X_i)) = G \cdot (|f^{maj}| - |f^{min}|) \cdot \hat{I}(x_i)$$

Since the real situation contains several unknown factors, failure to select appropriate robustness measures may lead to performance degradation or instability of the system [39]. The biggest advantage of the roulette algorithm is that it is able to carry more information for the few classes being sampled multiple times. It also improves the quality of the few newly generated class instances.

- **Generation of New Instances.**

    Based on the above operations, appropriate anchors are selected to generate new instances of higher quality. The generation steps based on Gaussian distribution are as follows:

    An arbitrary direction is chosen as the starting direction vector of the anchor instance $\hat{I}(X_i)$. The end point of direction training is denoted as $V = (v_1, v_2, \cdots, v_l)$. The details are shown in Fig 2 below.

    Here $X_i$, V and $X^t$ are all in the same check, The directional vector is given by the following formula:

$$\overrightarrow{X_iV} = \overrightarrow{OV} - \overrightarrow{OX_i}$$

Where O is the origin of the coordinates, $\overrightarrow{OV}$ and $\overrightarrow{OX_i}$ are the position vectors of points V and $X_i$.

Determine the Distance Between the New Instance and the Anchor Instance

According to the Gaussian distribution $W(\mu_i, \alpha\sigma_i)$, the mean $\mu_i$ is 0 and the standard deviation $\sigma_i$, $\alpha$ is the scaling factor:

$$\alpha_i = \frac{\sum_{k=1}^{l} |x_{ik} - x_{jk}|}{\sum_{k=1}^{i} (|x_{ik}| + |x_{jk}|)}$$

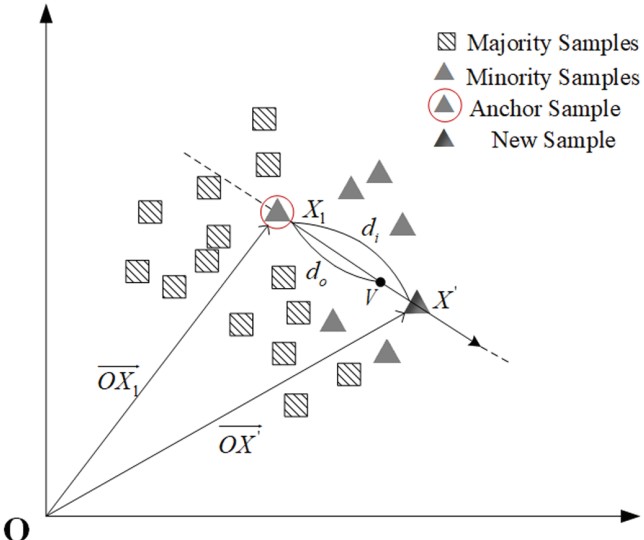

**Fig 2. Selection of instances.**

The distance between the new instance and the anchor instance is calculated. Also, calculate the scale factor between this distance and the length of the direction vector q:

$$q = \frac{\left|\overrightarrow{X_i X}\right|}{\left|\overrightarrow{X_i V}\right|}$$

In this, $d_0 = \left|\overrightarrow{X_i V}\right| = \sqrt{\sum_{k=1}^{L}(v_k - x_k)}$. So, the final new instance coordinates $X'$ can be expressed as:

$$X' = \{x_1 + q(v_1 - x_1), x_2 + q(v_2 - x_2), ..., x_l + q(v_l - x_l)\}$$

GDO generates new samples based on the statistical characteristics of the data, effectively alleviating the problems of limited data volume and unbalanced data classes. Compared with traditional oversampling methods such as SMOTE, GDO generates samples based on the statistical characteristics of the data, which can effectively reduce the influence of noise samples, enhance sample diversity, and thereby improve the performance of the model. The RBFN, discussed below, further improves the model's performance in classifying students' academic performance by virtue of its good nonlinear fitting and generalisation abilities.

## The module of RBFN

The RBFN model was first proposed by Moody and Darken as a typical feed-forward neural network. The output of a radial basis function network is a linear combination of the input radial basis function and neuron parameters. As shown in Fig 3, it is designed to overcome the limitations of traditional neural networks in dealing with complex nonlinear problems. By introducing a radial function as a non-linear activation function, RBFN is able to effectively capture potentially complex patterns and relationships in data. The model consists of an input layer, a radial function layer and an output layer to classify a standardized new dataset.

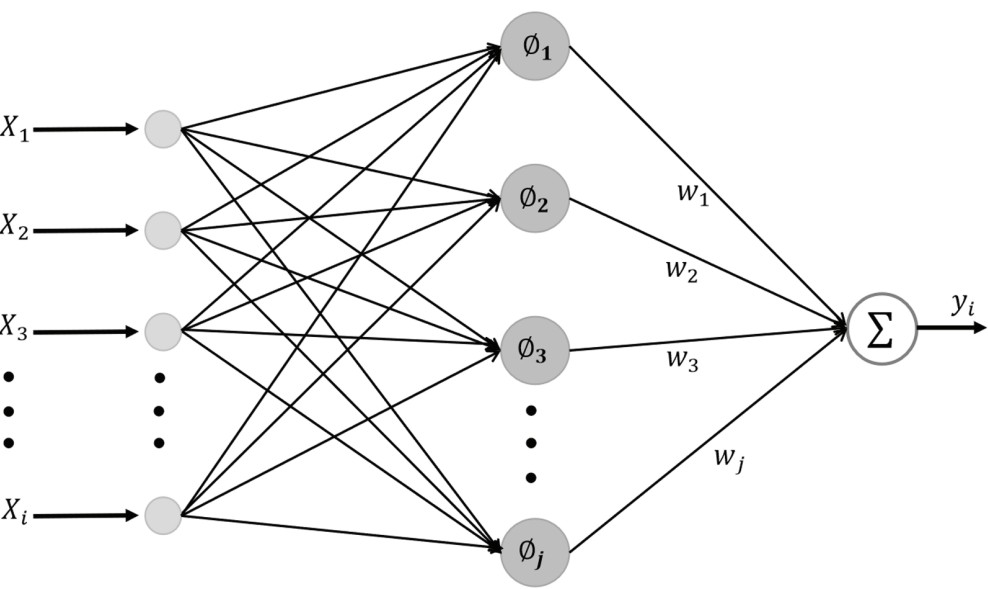

**Fig 3. The model of RBFN.**

- **Feature selection and Standardization.**

    We assigned the original features in the dataset in different combinations: overall features (all features), personal problems, family problems and educational habits. And these features were standardized with a mean of 0 and a variance of 1.

- **Radial Function Output Layer.**

    The radial basis function maps the input features to the higher space. Each of its radial functions $\phi_j$ is composed of a centre $\mu_j$ and a width parameter $\sigma_j$. The formulae are defined below:

$$\phi_j(x) = \exp\left(-\frac{\|x-\mu_j\|^2}{2\sigma_j^2}\right)$$

Where $\|x - \mu_j\|$ denotes the distance between the input sample x and the center $\mu_j$, and $\sigma_j$ is used to denote the width of the control basis function.

- **The Layer of Output.**

    The output is a weighted linear combination of the radial function's to get the final classification with the following formula:

$$y = \sum_{j=1}^{M} w_j \phi_j(x) + b$$

where y is the output, $w_j$ is the weight, b is the bias, and M is the number of radial functions.

The loss function is used to measure the difference between predicted and actual values. The classification function used for the experiment is the cross-entropy loss function:

$$L = -\frac{1}{N} \sum_{i=1}^{N} \sum_{c=1}^{C} y_i \log(\hat{y}_{i,c})$$

L is the value of the loss function, N is the number of samples, and C is the number of categories. $y_{i,c}$ is the actual value of i samples on category c, and $\hat{y}_{i,c}$ refers to the predicted value of i samples on category c.

Through the above steps, the radial basis function network can effectively classify student performance. The model tunes the model parameters by standardizing the input features, high-dimensional mapping of the radial basis function layer, linear combination of the output layers, and mini-mising the loss function. Thus, the model achieves accurate prediction of student performance.

## Experiment

In order to improve the running rate and performance of the model, when we set the learning rate lr of RBFN, the default values of the two datasets were 0.005, the embedding dimension was set to 32, and the number of neurons in the hidden layer was 300. For the GDO module, we adopt a neighbor number of 5, a covariance coefficient of 0.2, and a target oversampling rate that are dynamically adjusted according to different feature groups.

### Datasets

The datasets we used are all publicly available datasets, the data provider is the UCI database, which can be categorized into two types: the higher education student performance evaluation dataset and the student performance dataset, which can be downloaded from the links respectively: [https://archive.ics.uci.edu/dataset1] and [https://archive.ics.uci.edu/dataset2], and the characterizations of the different datasets are summarized in Tables 1 and 2. For ease of representation, we call the evaluation dataset and performance dataset in subsequent descriptions. This study places this dataset in the context of different machine learning and deep learning methods for predicting actual student achievement performance. By analyzing multiple characteristics of students provided in the dataset, such as personal information, academic performance, living habits and family background, among other factors. It aims to analyze and understand how these factors affect students' performance in academic achievement. This model improves teaching methods and educational decision-making based on data prediction model results.

The evaluation dataset has 146 data entries consisting of 32 features, of which 1–10 are individual problems, 11–16 are family problems, and the rest are educational habits. Student achievements are divided into eight categories: AA, BA, BB, CB, CC, DC, DD, and FF. The experiment classified them into three categories based on their performance characteristics: excellent in category A, qualified in category B, and unqualified in category C. In order to validate the effectiveness of GDO-RBFN, we similarly conducted experiments on the performance dataset, which contains 396 pieces of data with 32 features, where 1–12 are individual problems, 13–25 are family problems, and 26–32 are educational habits, which similarly categorize student performance.

### Evaluation indicators

In the field of education, accurately evaluating the performance of models is particularly important for classifying student grades. Confusion matrix provides a structured method to summarize the accuracy of model prediction results. Among them, there are true cases (TP), false negative cases (FN), false positive cases (FP), and true negative cases (TN), as shown in Table 3. It helps to better classify students' academic performance in research.

By using the following different indicators, more effective and accurate student intervention and support strategies can be developed based on the prediction results of different models. Ensure that educational resources can help students who truly need assistance to the maximum extent possible, and improve overall educational outcomes. The following is

**Table 1. Higher education students performance evaluation dataset feature description.**

| Number | Features | Feature description |
|---|---|---|
| 1 | Student Age | 1: 18-21, 2: 22-25, 3: 26 years old and above |
| 2 | Gender | 1: Female, 2: Male |
| 3 | High School Graduation Type | 1: Private High School, 2: State, 3: Others |
| 4 | Scholarship Type | 1: None, 2: 25%, 3: 50%, 4: 75%, 5: Full scholarship |
| 5 | Additional work | 1: Yes, 2: No |
| 6 | Regular artistic or sports activities | 1: Yes, 2: No |
| 7 | Do You Have a Partner | 1: Yes, 2: No |
| 8 | Total Salary (if any) | 1: $135–200, 2: $201–270, 3: $270–340, 4: $341–410, 5: Above $410 |
| 9 | Transportation to University | 1: Bus, 2: Private car/Taxi, 3: Bike, 4: Others |
| 10 | Types of Accommodation in Cyprus | 1: Renting house, 2: Dormitory, 3: Together with family, 4: Others |
| 11 | Mother's Education | 1: Primary school, 2: Middle school, 3: High school, 4: University, 5: Master's degree, 6: Doctoral degree |
| 12 | Father's Education | 1: Primary school, 2: Middle school, 3: High school, 4: University, 5: Master's degree, 6: Doctoral degree |
| 13 | Number of Sisters/Brothers (if any) | 1: 1, 2: 2, 3: 3, 4: 4, 5: 5 or above |
| 14 | Parents' Condition | 1: Married, 2: Divorce, 3: Death - one or both of them |
| 15 | Mother's profession | 1: Retired, 2: Housewives, 3: Government Officials, 4: Private Sector Employees, 5: Self Employed, 6: Others |
| 16 | Father's profession | 1: Retired, 2: Housewives, 3: Government Officials, 4: Private Sector Employees, 5: Self Employed, 6: Others |
| 17 | Weekly Study Time | 1: None, 2: <5 hours, 3: 6–10 hours, 4: 11–20 hours, 5: 20 hours or more |
| 18 | Reading Frequency (Non Scientific Books/Journals) | 1: None, 2: Sometimes, 3: Often |
| 19 | Reading Frequency (Scientific Books/Journals) | 1: None, 2: Sometimes, 3: Often |
| 20 | Attend Seminars/Conferences Related to the Department | 1: Yes, 2: No |
| 21 | The Impact of Your Project/Activity on Success | 1: Positive, 2: Negative, 3: Neutral |
| 22 | Attend Class | 1: Always, 2: Sometimes, 3: Never |
| 23 | Mid Term Exam Preparation 1 | 1: Alone, 2: With friends, 3: Not applicable |
| 24 | Mid Term Exam Preparation 2 | 1: The closest date to the exam, 2: Regular semester, 3: Never |
| 25 | Take notes in class | 1: Never, 2: Sometimes, 3: Always |
| 26 | Classroom Listening | 1: Never, 2: Sometimes, 3: Always |
| 27 | Whether to Increase Interest and Success in the Course | 1: Never, 2: Sometimes, 3: Always |
| 28 | Flipped Classroom | 1: Useless, 2: Useful, 3: Not applicable |
| 29 | Cumulative GPA of last semester (/4.00) | 1: <2.00, 2: 2.00–2.49, 3: 2.50–2.99, 4: 3.00–3.49, 5: 3.49 or above |
| 30 | Expected Cumulative GPA during Graduation Period (/4.00) | 1: <2.00, 2: 2.00–2.49, 3: 2.50–2.99, 4: 3.00–3.49, 5: 3.49 or above |
| 31 | Course ID | |
| 32 | Output Level | 0: Fail, 1: DD, 2: DC, 3: CC, 4: CB, 5: BB, 6: BA, 7: AA |

**Note:** https://archive.ics.uci.edu/dataset/856/higher+education+students+performance+evaluation

the formula for using indicators and the purpose of selecting them as experimental evaluation indicators. Indicator calculation: Use the weighted average of Acc, Precision, Recall, and F1 as the final measurement indicator. Specifically, the accuracy, recall, and F1 score of each category will be weighted based on their proportion in the dataset. Then, sum up to obtain the weighted average of the entire dataset. This ensures that each category is treated fairly in the evaluation, thereby more accurately reflecting the performance of the model. Accuracy, calculated using the following formula:

- **Logistic Regression(LR):** The classic linear classification model, commonly used in the mid-20th century, is mainly applied to solve binary classification problems. Its core principle maps features between 0-1 probabilities through linear models and sigmoid functions.

**Table 2. Student performance dataset feature description.**

| Number | Features | Feature description |
|---|---|---|
| 1 | Student's school | Binary: 'GP' - gabriel pereira or 'MS' - mousinho da silveira |
| 2 | Student's sex | Binary: 'F' - female or 'M' - male |
| 3 | Student's age | Numeric: from 15 to 22 |
| 4 | Student's home address | Binary: 'U' - urban or 'R' - rural |
| 5 | Family size | Binary: 'LE3' - less or equal to 3 or 'GT3' - greater than 3 |
| 6 | Parent's cohabitation status | Binary: 'T' - living together or 'A' - apart |
| 7 | Mother's education | Numeric: 0 - none, 1 - primary education, 2 - secondary education, 3 - higher education |
| 8 | Father's education | Numeric: 0 - none, 1 - primary education, 2 - secondary education, 3 - higher education |
| 9 | Mother's job | Nominal: 'teacher', 'health' care related, civil 'services', 'at home' or 'other' |
| 10 | Father's job | nominal: 'teacher', 'health' care related, civil 'services', 'at home' or 'other' |
| 11 | Reason to choose this school | nominal: close to 'home', school 'reputation', 'course' preference or 'other' |
| 12 | Student's guardian | nominal: 'mother', 'father' or 'other' |
| 13 | Home to school travel time | numeric: 1 - <15 min., 2 - 15 to 30 min., 3 - 30 min. to 1 hour, or 4 - >1 hour |
| 14 | Weekly study time | numeric: 1 - <2 hours, 2 - 2 to 5 hours, 3 - 5 to 10 hours, or 4 - >10 hours |
| 15 | Number of past class failures | numeric: n if 1<=n<3, else 4 |
| 16 | Extra educational support | Binary: yes or no |
| 17 | Family educational support | Binary: yes or no |
| 18 | Extra paid classes within the course subject | Binary: yes or no |
| 19 | Extra-curricular activities | Binary: yes or no |
| 20 | Attended nursery school | Binary: yes or no |
| 21 | Wants to take higher education | Binary: yes or no |
| 22 | Internet access at home | Binary: yes or no |
| 23 | With a romantic relationship | Binary: yes or no |
| 24 | quality of family relationships | numeric: from 1 - very bad to 5 - excellent |
| 25 | Free time after school | Numeric: from 1 - very low to 5 - very high |
| 26 | Going out with friends | Numeric: from 1 - very low to 5 - very high |
| 27 | Workday alcohol consumption | Numeric: from 1 - very low to 5 - very high |
| 28 | Weekend alcohol consumption | Numeric: from 1 - very low to 5 - very high |
| 29 | Current health status | Numeric: from 1 - very bad to 5 - very good |
| 30 | Number of school absences | numeric: from 0 to 93 |
| 31 | First and second period grade | Numeric: from 0 to 20 |
| 32 | Final grade | numeric: from 0 to 20, output target |

**Note:** https://archive.ics.uci.edu/dataset/320/student+performance

**Table 3. Confusion matrix.**

| | Predicted | | |
|---|---|---|---|
| **Reference** | Positive and Negative | Positive | Negative |
| | Positive | True Positive ( TP) | False Negative ( FN) |
| | Negative | False Positive (FP) | True Negative (TN) |

- **Support Vector Machine(SVM):** Originating from the supervised algorithm proposed by Vladimir Vapnik and Corinna Cortes in the 1990s. It is suitable for linear and nonlinear classification, regression, and anomaly detection tasks, and performs classification by maximizing the inter class interval.
- **Random Forest(RF):** The ensemble learning algorithm proposed by Leo Breiman in 2001 utilizes multiple decision trees for ensemble learning. It improves the accuracy and robustness of the model through voting or averaging predicted values.
- **Multilayer Perceptron(MLP):** As early as the 1980s, it was widely studied and promoted as a classic feedforward neural network structure. It is composed of multiple neurons, each fully connected to the next layer.

- **Artificial Neural Network(ANN):** This is a relatively ancient concept, but the development and application of modern neural networks have significantly increased since the 1990s. It simulates the neural network of the human brain and is used to solve various machine learning problems.
- **Bidirectional Long Short-Term Memory(BILSTM):** Deep learning models, proposed by Schuster and Paliwal in 1997.They are used for processing sequential data. They consider past and future contextual information at every time step.
- **Radial Basis Function Network(RBFN):** The classic neural network structure was widely studied and applied in the late 1980s and early 1990s. Its hidden layer uses radial basis functions as activation functions, typically used for regression and classification problems.

## Results

We predict students' grades based on various characteristics according to the experimental content. The experiment can be roughly divided into two parts: overall feature classification and data classification based on GDO amplification. The overall feature classification includes the classification of personal issues, family issues, and educational habits. In order to further enhance the generalisation ability of the model and validate its applicability across different cohorts, a 10-fold cross-validation was carried out during the course of the study, specifically, the original dataset was partitioned into 10 mutually exclusive subsets, and each time, 9 of them were used as the training set, and the remaining 1 subset as the test set. The process was repeated 10 times to ensure that each sample had the opportunity to participate in the evaluation as a test sample. The experiments were based on pytorch to build the RBFN model and Adam was used for learning optimisation.

### Comparison of pre-amplification models

In this experiment, three machine learning models (LR, SVM, RF) and four deep learning models (MLP, ANN, BILSTM, RBFN) were used to classify the data, and the model performance was evaluated based on accuracy, precision, recall, and F1 score. The results of the categorization by overall characteristics are given in Table 4.

By classifying based on overall features, i.e. using dataset 32 features. The experiment uses machine learning LR, SVM, RF, and deep learning MLP, ANN, BILSTM, and RBFN models. Among all models, RF performs the best in classification, with higher accuracy, recall, and F1 score than other models. This indicates that RF has significant advantages in processing high-dimensional and complex datasets, as its model can reduce overfitting and has good robustness to outliers. The LR model achieves the best accuracy, rarely mistaking high achieving students for local grades. Although it has good processing ability on linearly separable

**Table 4. Classify by overall characteristics. The best results are in bold and the second best results are in italics (%).**

| Model | | Evaluation dataset | | | | Performance dataset | | | |
|---|---|---|---|---|---|---|---|---|---|
| | | Accuracy | Precision | Recall | F1-scores | Accuracy | Precision | Recall | F1-scores |
| **ML** | **LR** | 78.05 | **73.10** | 78.05 | 75.23 | 82.50 | *84.39* | 82.50 | 81.29 |
| | **SVM** | 73.81 | *72.61* | 73.81 | 72.45 | 81.37 | 70.06 | 81.37 | 75.10 |
| | **RF** | **82.86** | 71.60 | **82.86** | **76.60** | *82.85* | 78.73 | *82.85* | 75.15 |
| **DL** | **MLP** | 78.76 | 69.42 | 78.76 | 73.67 | 74.57 | 71.76 | 74.57 | 72.83 |
| | **ANN** | 80.10 | 70.18 | 80.10 | 74.69 | 82.22 | 83.00 | 82.22 | *81.71* |
| | **BILSTM** | 78.71 | 69.46 | 78.71 | 73.67 | 81.94 | 82.66 | 81.94 | 80.78 |
| | **RBFN** | *81.52* | 71.09 | *81.52* | *75.73* | **92.78** | **92.84** | **92.78** | **92.58** |

data, it has limitations in processing complex and nonlinear features. SVM has natural advantages in handling small samples and high-dimensional data, but based on experimental data, there may be too many non-linear features, and the actual effect is not ideal. MLP can capture nonlinear features in data through the connection of multiple layers of neurons, but its performance is slightly lower than RF and RBFN. Compared to MLP, ANN has a slight advantage in nonlinear feature processing, but its actual performance is still inferior to RF. BILSTM can capture bidirectional dependencies in sequence data, but in this experiment, its performance did not significantly surpass other models. RBFN is second only to RBFN in other indicators, demonstrating its potential in capturing non-linear features of data.

Describing individual problems based on 1-10 features of the overall characteristics, as shown in Table 5, except for logistic regression, the results of all other models run show a significant decrease compared to the overall characteristics. Among them, compared to the overall features, the best method for classifying individual problem features is also RF, with an accuracy decrease of 2.86%, an accuracy increase of 0.51%, a recall decrease of 2.6%, and F1 scores decrease of 1.09%. This indicates that when the number of features decreases, especially when key family issues and educational habits are lacking, the decision tree construction and overall performance of the model will be affected. The indicators of RBFN have also decreased by about 3%. When key features are missing, the model's ability to capture data complexity decreases, leading to a decrease in its classification performance. Only LR is not affected like other class models, indicating that logistic regression performs well in handling linearly separable data and still has strong classification ability for personal problem features. Despite the lack of family issues and educational habits, their overall performance is not significantly affected.

The 11–16 features in the overall characteristics describe family problems, and the impact of family feature classification on each model varies, as shown in Table 6. The best performing

**Table 5. Classify by personal issues. The best results are in bold and the second best results are in italics (%).**

| Model | | Evaluation dataset | | | | Performance dataset | | | |
|---|---|---|---|---|---|---|---|---|---|
| | | Accuracy | Precision | Recall | F1-scores | Accuracy | Precision | Recall | F1-scores |
| ML | LR | **80.00** | 68.34 | **80.00** | *73.64* | 81.52 | 83.25 | 81.52 | *85.29* |
| | SVM | 73.14 | 68.28 | 73.14 | 70.56 | 78.37 | 72.06 | 78.37 | 74.30 |
| | RF | **80.00** | **72.11** | **80.00** | **75.51** | *83.75* | 75.23 | *83.75* | 77.35 |
| DL | MLP | 77.29 | *68.77* | 77.29 | 72.63 | 79.59 | 70.76 | 79.59 | 75.93 |
| | ANN | 77.29 | 68.70 | 77.29 | 72.51 | 80.12 | 84.08 | 80.12 | 81.54 |
| | BILSTM | 77.24 | 67.93 | 77.24 | 72.19 | 82.94 | *85.66* | 82.94 | 84.78 |
| | RBFN | 78.00 | 68.02 | 78.00 | 72.61 | **92.78** | **92.84** | **92.78** | **92.58** |

**Table 6. Classified by family issues. The best results are in bold and the second best results are in italics (%).**

| Model | | Evaluation dataset | | | | Performance dataset | | | |
|---|---|---|---|---|---|---|---|---|---|
| | | Accuracy | Precision | Recall | F1-scores | Accuracy | Precision | Recall | F1-scores |
| ML | LR | **80.76** | 68.42 | **80.76** | *74.00* | 81.14 | 79.16 | 81.14 | *79.36* |
| | SVM | 77.38 | 68.68 | 77.38 | 72.63 | 77.10 | 78.37 | 77.10 | 75.08 |
| | RF | 77.48 | **69.37** | 77.48 | 73.09 | 79.75 | 75.02 | 78.75 | 76.24 |
| DL | MLP | 80.10 | 68.90 | 80.10 | 73.98 | 79.87 | 1,0.643,0 79.76 | 79.87 | 78.93 |
| | ANN | 79.48 | *69.25* | 79.48 | 73.94 | 80.33 | 74.30 | 80.33 | 76.90 |
| | BILSTM | 79.38 | 68.86 | 79.38 | 73.51 | *81.17* | 71.45 | *81.17* | 72.66 |
| | RBFN | *80.19* | 68.73 | *80.19* | **74.02** | **88.37** | **89.17** | **88.37** | **87.38** |

models are LR and RBFN. Among them, RF and RBFN are more sensitive to changes in feature sets, and when only using household features, there are obvious signs of decline in accuracy and other indicators. Especially in the case of household characteristics, the effectiveness of RF in capturing information is not as good as in the case of overall characteristics, and the effect decreases significantly. LR has slightly improved in accuracy and precision, while the rest has decreased, indicating that family characteristics play an important role in capturing potential patterns of student performance in LR models, but at a slight sacrifice of accuracy. ANN, BILSTM, SVM and other models have an impact on classification performance, but the effect is not significant.

By categorizing the last 16 features of the overall characteristics as educational habit features, as shown in Table 7, it is not difficult to see that compared to the overall features, using only the last 16 features significantly improves the accuracy of predicting classification grades, indicating the crucial role of educational habit features in grade classification. Educational habits can effectively capture potential differences in academic performance, including course interest, study time, exam preparation, etc. These factors have a significant impact on student performance, and can directly reflect students' learning ability and academic level from these instance differences. According to the experimental results, the best model for classifying educational habits is RBFN, which improves accuracy by 2.7%, precision by 5.6%, recall by 2.71%, and F1 scores by 4.15%. The RBFN model performs well in handling high-dimensional data and nonlinear relationships, and therefore shows significant improvement when using educational habit features. The limited increase in LR and RF models indicates that educational habit features can improve accuracy, but the effect is minimal.

## Comparison of amplified models

The three types of data used in the experiment have levels A, B, and C. Due to the limited number of 145 entries in the original dataset. Among them, the most common category is 120 cases, and the remaining two categories will be amplified in the experiment. Mainly utilizing oversampling based on Gaussian distribution, generating new sample nodes using Gaussian distribution to provide sufficient data for model training, achieving more accurate classification of student grades. The amplification was categorized by overall characteristics as shown in Table 8.

After Gaussian distribution based oversampling (GDO) amplification, each model showed significant improvements in accuracy, precision, recall, and F1 score. Specifically, the performance indicators of LR, SVM, RF, MLP, ANN, BILSTM, and RBFN models after amplification have improved by nearly 10% compared to before amplification. Especially the RBFN model, it performs the best in all indicators, with an accuracy of 95.00%, precision of 95.52%, recall rate of 95.00%, and F1 score of 94.95%. RBFN exhibits extremely high performance

**Table 7. Classified by educational habits. The best results are in bold and the second best results are in italics (%).**

| Model | | Evaluation dataset | | | | Performance dataset | | | |
|---|---|---|---|---|---|---|---|---|---|
| | | Accuracy | Precision | Recall | F1-scores | Accuracy | Precision | Recall | F1-scores |
| ML | LR | 79.38 | 70.95 | 79.38 | 74.82 | 80.14 | 76.34 | 80.14 | 77.82 |
| | SVM | 80.71 | *76.04* | 80.71 | 77.85 | 81.59 | 79.26 | 81.59 | 79.88 |
| | RF | *82.81* | 72.11 | *82.81* | 76.73 | 82.10 | 74.27 | 82.10 | 78.45 |
| DL | MLP | 82.24 | 74.70 | 82.24 | *78.04* | 85.55 | 78.42 | *85.55* | 80.26 |
| | ANN | 82.19 | 74.74 | 82.19 | 77.97 | 81.38 | *82.66* | 81.38 | *80.67* |
| | BILSTM | 80.14 | 74.20 | a80.14 | 76.75 | 81.85 | 75.26 | 81.85 | 78.39 |
| | RBFN | **84.23** | **76.76** | **84.23** | **79.88** | **86.59** | **83.22** | **86.59** | **85.03** |

**Table 8. Classify according to overall characteristics after amplification. The best results are in bold and the second best results are in italics (%).**

| Model | | Evaluation dataset | | | | Performance dataset | | | |
|---|---|---|---|---|---|---|---|---|---|
| | | Accuracy | Precision | Recall | F1-scores | Accuracy | Precision | Recall | F1-scores |
| ML | LR | 91.71 | 92.96 | 91.71 | 91.46 | 92.48 | 94.26 | 92.48 | 92.73 |
| | SVM | 92.53 | 93.38 | 92.53 | 92.29 | 91.38 | 92.56 | 91.38 | 92.47 |
| | RF | 93.63 | *94.99* | 93.63 | 93.69 | 93.22 | *95.12* | 93.22 | *93.67* |
| DL | MLP | 94.18 | 94.85 | 94.18 | *94.09* | 88.57 | 95.06 | 88.57 | 91.67 |
| | ANN | *94.19* | 94.66 | *94.19* | 94.06 | *93.88* | 94.69 | *93.88* | 93.46 |
| | BILSTM | 93.36 | 94.05 | 93.36 | 93.18 | 91.94 | 92.52 | 91.94 | 91.25 |
| | RBFN | **95.00** | **95.52** | **95.00** | **94.95** | **97.56** | **95.56** | **97.56** | **94.28** |

in adapting to GDO, far exceeding the optimal RF model before amplification. This phenomenon indicates that RBFN has certain requirements for data training volume, and sufficient training volume can further capture potential feature relationships. In addition, GDO amplification provides more authentic data, weakening the negative impact of class imbalance on model performance and improving the actual performance of the model in overall classification tasks. This further proves the effectiveness of GDO method in solving data imbalance problems and the powerful feature capture ability of RBFN model under sufficient training data.

Predicting student performance classification based on the top 10 individual questions according to overall features significantly improved the model's classification ability on individual question features, especially with SVM, RF, MLP, and RBFN showing the most significant improvement, as shown in Table 9. This indicates that through GDO amplification, not only does it increase the sample size, but it also enhances the model's ability to capture and learn complex features, allowing the model to still perform well in classifying data with fewer features. There is a significant decrease in LR, as the performance of the LR model is affected when the newly generated data samples do not fully conform to the linear distribution of the original data. The ability of LR to process data is also limited by the increase in data complexity and the number of nonlinear features.

According to the classification of family problem characteristics, the impact of family problem characteristics on various models has significantly improved after data augmentation, as shown in Table 10. The RF model showed significant improvement in all indicators, with an accuracy of 92.22%. This indicates that the RF model can better capture data features and improve classification performance in terms of family problem characteristics. The RBFN model also performed well, with an accuracy improvement of 88.06%, demonstrating its advantages in capturing data complexity and feature relationships. In contrast, the LR model shows a slight decrease in accuracy and precision, indicating that family features have to some

**Table 9. Categorize by personal questions after amplification. The best results are in bold and the second best results are in italics (%).**

| Model | | Evaluation dataset | | | | Performance dataset | | | |
|---|---|---|---|---|---|---|---|---|---|
| | | Accuracy | Precision | Recall | F1-scores | Accuracy | Precision | Recall | F1-scores |
| ML | LR | 77.23 | 77.01 | 77.23 | 76.10 | 77.09 | 76.74 | 72.09 | 78.09 |
| | SVM | 87.50 | 88.76 | 87.50 | 86.93 | 85.69 | 86.93 | 85.69 | 84.71 |
| | RF | **91.11** | *91.76* | **91.11** | **91.12** | 88.16 | 89.34 | 88.16 | *90.42* |
| DL | MLP | 88.33 | 89.37 | 88.33 | 87.84 | *89.52* | 88.05 | *89.52* | 90.10 |
| | ANN | 87.78 | 88.75 | 87.78 | 87.29 | 85.06 | 86.40 | 85.06 | 86.26 |
| | BILSTM | 87.79 | 88.80 | 87.79 | 82.27 | 86.09 | *89.71* | 86.09 | 89.04 |
| | RBFN | *90.83* | **91.97** | *90.83* | *90.37* | **91.25** | **90.27** | **91.25** | **92.03** |

**Table 10. Categorize by family issues after amplification. The best results are in bold and the second best results are in italics (%).**

| Model | | Evaluation dataset | | | | Performance dataset | | | |
|---|---|---|---|---|---|---|---|---|---|
| | | Accuracy | Precision | Recall | F1-scores | Accuracy | Precision | Recall | F1-scores |
| ML | LR | 77.75 | 78.45 | 77.75 | 77.57 | 79.06 | 78.00 | 79.06 | 79.50 |
| | SVM | 80.56 | 81.69 | 80.56 | 79.36 | 85.16 | 86.34 | 85.16 | 84.60 |
| | RF | **92.22** | **93.21** | **92.22** | **92.19** | **93.27** | **93.88** | **93.27** | **93.00** |
| DL | MLP | 83.61 | 84.73 | 83.61 | 82.91 | 84.62 | 85.25 | 84.62 | 83.21 |
| | ANN | 83.33 | 84.46 | 83.33 | 82.66 | 83.00 | 84.22 | 83.00 | 83.27 |
| | BILSTM | 82.23 | 83.07 | 82.23 | 81.04 | 84.52 | 85.39 | 84.52 | 84.10 |
| | RBFN | *88.06* | *89.08* | *88.06* | *87.70* | *89.37* | *89.45* | *89.37* | *88.00* |

extent affected its classification ability. Models such as SVM, MLP, ANN, and BILSTM have all shown improvements in classification performance, although the magnitude is not as significant as RF and RBFN, they still perform stably. This indicates that the data provided by GDO amplification effectively improves the classification performance of family problem features and enhances the overall performance of various models in this feature classification.

According to the classification of educational habit characteristics, after data GDO amplification, each model showed significant improvement in classification performance when using educational habit characteristics, as shown in Table 11. Among them, the RBFN model performs excellently, with an accuracy rate of up to 93.58%, precision of 94.6%, recall rate of 93.58%, and F1 Score of 92.92%, demonstrating the advantages of this model in high-dimensional feature and nonlinear relationship data processing. The performance improvement of SVM and other models is also quite significant, especially in accuracy and precision, which highlights the necessity of using GDO to amplify data. The improvement of the retrospective model is relatively limited, and this model is better applied to linear classification. After GDO amplification, the potential data capture ability of RBFN is effectively amplified.

In the above experiment, regardless of whether GDO amplification is used or not, the most effective methods for classifying student performance are based on overall features and educational features. However, after using GDO amplification, various methods showed significant improvements, especially the RBFN model, which showed obvious superiority in handling nonlinear relationships and high-dimensional features. In terms of data, GDO amplification can improve the minority class data to a certain extent, and sufficient training can better capture the relationship between potential features of the data, alleviating the problem of class imbalance. In summary, the GDP-RBFN adopted has the most ideal performance in score classification.

**Table 11. Expand and classify according to individual educational habits. The best results are in bold and the second best results are in italics (%).**

| Model | | Evaluation dataset | | | | Performance dataset | | | |
|---|---|---|---|---|---|---|---|---|---|
| | | Accuracy | Precision | Recall | F1-scores | Accuracy | Precision | Recall | F1-scores |
| ML | LR | 77.40 | 77.71 | 77.40 | 76.95 | 80.71 | 81.66 | 80.71 | 82.30 |
| | SVM | 90.88 | 92.29 | 90.88 | 90.43 | 90.12 | 93.05 | 90.12 | 91.40 |
| | RF | 92.92 | 93.92 | 92.92 | *92.99* | *94.26* | *94.71* | *94.26* | 94.21 |
| DL | MLP | 92.80 | 93.84 | 92.80 | 92.50 | 93.20 | 94.50 | 93.20 | *94.78* |
| | ANN | 92.52 | 93.62 | 92.52 | 92.16 | 92.10 | 93.64 | 92.10 | 92.39 |
| | BILSTM | *93.10* | *94.14* | *93.10* | 92.78 | 94.07 | 91.89 | 94.07 | 92.00 |
| | RBFN | **94.12** | **94.60** | **94.12** | **94.46** | **95.20** | **95.87** | **95.20** | **95.67** |

## Discussion

### Reliability of synthetic data

Since the data were synthesised using GDO, we conducted experiments in order to verify the credibility of the synthesised data, using the Homogeneity of Variance, P-Value, distribution overlap and Fréchette distance as evaluation indicators, and the results of the experiments are shown in Table 12. Starting from the Homogeneity of Variance and P-Value, the Homogeneity of Variance and P-Value of the four categories are higher than 0.05, with the P-Value of the overall dataset being 0.8387, personal problems 0.9210, family problems 0.9075, and educational habits 0.9125. This indicates that there is no significant difference in the homogeneity of variance of the data in different categories, and the hypothesis of homogeneity of variance is valid. significant difference and the assumption of variance chi-square is valid. In terms of distribution overlap and Frechette distance, the distribution overlap values of the four categories are distributed between 0.94 and 0.97, close to 1, indicating that the distributions of raw and synthetic data on each feature are highly overlapped. And the Frechette distance is distributed between 0.02–0.04, all less than 0.05, indicating that the distribution of synthetic data and original data is very similar, suggesting that the synthetic data is of high quality. In summary, the distributions of the synthetic data on these categories were consistent with the original data with high confidence, further verifying the effectiveness of the GDO amplification method in generating credible synthetic data.

### Sensitivity parameter analysis

Fig 4 shows the performance metrics (accuracy, precision, and F1 score) of the model under different feature combinations as a function of oversampling rate p. When integrating all features, as the oversampling rate increases, various performance indicators gradually improve, reaching the highest value at an oversampling rate of 1, indicating a significant improvement in model performance with moderate oversampling. When using personal problem features, the accuracy and F1 score significantly decrease at an oversampling rate of 0.2, but gradually increase thereafter, reaching optimal results at 0.8, indicating that personal problem features are more sensitive to low oversampling rates. When applying family problem features, the performance of the model significantly decreases at low oversampling rates (0.2), but gradually improves with increasing oversampling rates and reaches optimal results at 1. When using educational habit features, the performance index slightly decreases at an oversampling rate of 0.6, but reaches its optimal effect at 0.8, indicating that educational habit features are sensitive to moderate oversampling. Overall, different feature combinations have varying sensitivities to oversampling rates, but moderate oversampling (typically between 0.6 and 1) can significantly improve model performance.

k denotes the number of nearest neighbours considered in calculating the weights of the minority class samples. the choice of the k value has a significant impact on the calculation of the weights of the minority class samples. Smaller values of k may lead to overconcentration

**Table 12. Reliability verification of synthetic data.**

|  | Homogeneity of variance | P-value | Distribution overlap | Fréchet distance |
|---|---|---|---|---|
| **Overall dataset** | 0.9177 | 0.8387 | 0.9512 | 0.03 |
| **Personal problem** | 0.9658 | 0.9210 | 0.9705 | 0.02 |
| **Family issues** | 0.9516 | 0.9075 | 0.9400 | 0.04 |
| **Educational habits** | 0.9529 | 0.9125 | 0.9617 | 0.03 |

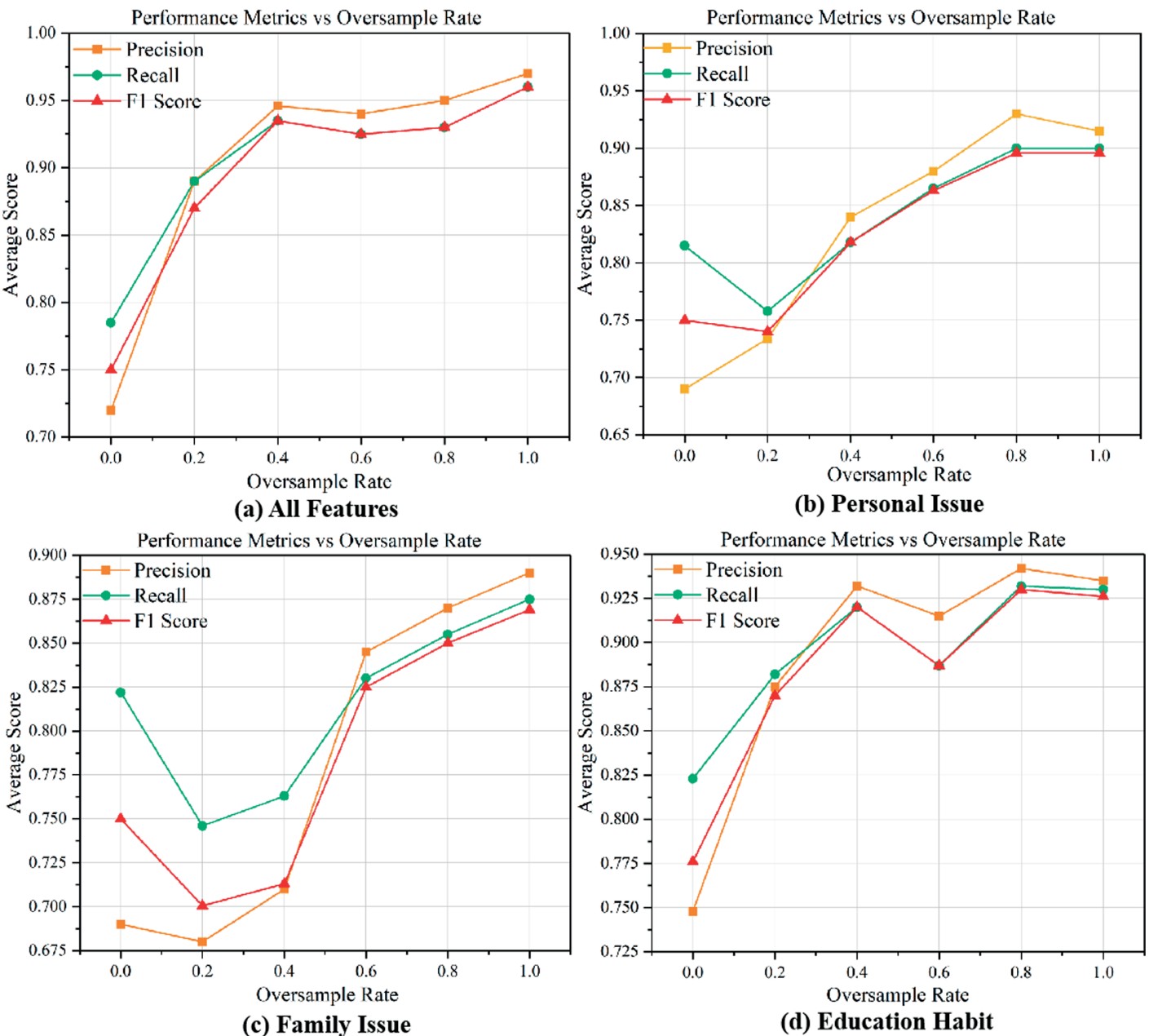

**Fig 4. GDO-RBFN is affected by oversampling rate.**

of the generated samples, while larger values of k may introduce noise. The following are the results of the comparison experiments.

The experimental comparison in Table 13 reveals that when the value of k is 5, all the evaluation indexes of educational habit feature classification are optimal for both datasets. We can optimise the performance of the GDO method by adjusting the size of k.12pt

**Table 13. Effect of k-value in GDO on the accuracy of classification of educational habitus features, and the best results are in bold (%).**

| Model | k | Higher education students performance evaluation | | | | Student performance | | | |
|---|---|---|---|---|---|---|---|---|---|
| | | Accuracy | Precision | Recall | F1-scores | Accuracy | Precision | Recall | F1-scores |
| RBFN | k = 4 | 92.16 | 93.65 | 93.14 | 92.37 | 94.24 | 93.22 | 93.24 | 92.11 |
| | k = 5 | **94.12** | **94.60** | **94.12** | **94.46** | **95.20** | **95.87** | **95.20** | **95.67** |
| | k = 6 | 93.14 | 94.22 | 93.06 | 91.75 | 94.88 | 95.06 | 94.85 | 91.39 |

## Limitations

Through the exploration in Tables 4–8, we found that the research results of grouping and classifying according to different features revealed that the RBFN method we proposed did not lead in all datasets and evaluation metrics. This might be due to its weak local sensitivity and adaptability to high-dimensional data, resulting in not leading comprehensively in all scenarios. The data characteristics also have the factor of insufficient matching degree with the local kernel function of RBFN. The fact that we adopted hyperparameters without optimizing for different datasets might also be one of the reasons why RBFN has not achieved comprehensive leadership.

Secondly, combining Tables 6 and 10, the ranking of our GDO-RBFN decreased instead after being augmented by GDO. Although GDO generates new samples through Gaussian distribution, the generated samples may still have certain similarities, resulting in insufficient sample diversity. This may affect the model's generalization ability for minority class samples. Furthermore, the synthetic samples generated by GDO may deviate from the true distribution characteristics of the original data, causing the model to learn false patterns on the enhanced dataset, which instead reduces the generalization ability on the original test set. This is particularly sensitive in educational data, where student characteristics often have complex underlying structures.

In the future, we will enhance the ability of RBFN to capture global features by introducing an attention mechanism or hierarchical radial basis functions, and use adaptive parameter optimisation methods to improve the model's adaptability to different data distributions. In addition, the loss function is customised and optimised for the specific needs of educational scenarios. For the selection of data augmentation strategy, we propose to adopt the conditional augmentation strategy to control the generation range and intensity of augmented data, perturb only the non-critical features, and retain the core features to carry out the next step of the study, so as to make my model more suitable for the application of educational classification models.

## Conclusion

The GDO-RBFN method, first proposed in the study, utilizes Gaussian distribution based data augmentation techniques combined with the RBFN model for student performance classification, effectively alleviating the impact of limited data volume and class imbalance. This method has shown significant potential and advantages in processing complex feature data, especially in the ability to handle non-linear relationships and multidimensional features, and can effectively predict the grades of classified students. However, when applied in small datasets or specific populations, the actual effectiveness of GDO-RBFN is limited by the quality and quantity of the data. The generalization ability and adaptability of the model need to be strengthened in order to adapt to the universality and effectiveness of different educational backgrounds and specific student groups. There is potential for application in feature engineering and integration technology in the future, and we will further expand this research.

In addition, exploring more deep learning methods to uncover deep patterns and correlated features in student performance prediction to address the challenges of data complexity and diversity. In the future, we will develop more educational data models to provide educators with more accurate analysis tools, help them better adjust students' learning strategies, and ultimately improve the quality of education, achieving personalized and overall optimization goals.

## Author contributions

**Conceptualization:** Ruishuang Sun.

**Data curation:** Ruishuang Sun.

**Formal analysis:** Ruishuang Sun.

**Funding acquisition:** Jianwei Dong.

**Investigation:** Jianwei Dong.

**Methodology:** Zhipeng Yan.

**Project administration:** Zhipeng Yan.

**Resources:** Zhipeng Yan.

**Supervision:** Xinyu Bi.

**Validation:** Xinyu Bi.

**Writing – original draft:** Meilun Shi.

**Writing – review & editing:** Meilun Shi.

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
