## [Decision Letter · Decision Letter 0]

15 Apr 2025

PONE-D-25-15254Research on Learning Achievement Classification Based on Machine LearningPLOS ONE

Dear Dr. Ruishuang,

Thank you for submitting your manuscript to PLOS ONE. After careful consideration, we feel that it has merit but does not fully meet PLOS ONE’s publication criteria as it currently stands. Therefore, we invite you to submit a revised version of the manuscript that addresses the points raised during the review process.

**ACADEMIC EDITOR: **Revise your submission as per the reviewers' comments.

We look forward to receiving your revised manuscript.

Kind regards,

Dr. Rajesh Kumar

Academic Editor

PLOS ONE

Reviewers' comments:

Reviewer's Responses to Questions

**Comments to the Author**

1. Is the manuscript technically sound, and do the data support the conclusions?

Reviewer #1: Partly

Reviewer #2: Yes

2. Has the statistical analysis been performed appropriately and rigorously? 

Reviewer #1: I Don't Know

Reviewer #2: Yes

3. Have the authors made all data underlying the findings in their manuscript fully available?

Reviewer #1: Yes

Reviewer #2: Yes

4. Is the manuscript presented in an intelligible fashion and written in standard English?

Reviewer #1: Yes

Reviewer #2: Yes

5. Review Comments to the Author

Reviewer #1: 1.The study tackles an important and socially impactful problem—predicting student academic performance using machine learning, which has growing significance in the era of data-driven education.

2.The proposed Gaussian Distribution-based Data Augmentation (GDO) technique coupled with a Radial Basis Function Network (RBFN) is interesting. However, is GDO a novel method or an adaptation of existing Gaussian-based oversampling techniques like SMOTE variants?

3.The reported accuracy (94.12%) and F1 score (94.46%) are impressive. How does this compare with other baseline models without GDO? Were these improvements statistically significant?

4.It is suggested to add article entitled “Al-Ali et al. Analyzing Socio-Academic Factors and Predictive Modeling of Student Performance Using Machine Learning Techniques” to the literature review.

5.While the classification accuracy is emphasized, does the study address interpretability of the models? Which features were most influential in predicting academic achievement, and can these be translated into actionable insights for educators?

6.The paper mentions using variance homogeneity and p-values to evaluate the synthetic data quality. Could more advanced metrics like distribution overlap, Fréchet distance, or t-SNE visualizations be considered for assessing augmentation effectiveness?

7.Did the original dataset suffer from class imbalance (e.g., high vs. low-performing students)? How does GDO compare to traditional oversampling techniques like SMOTE, ADASYN, or GAN-based augmentations?

8.The paper acknowledges the limitation in generalization to small datasets or specific student populations. Has cross-validation across multiple cohorts or schools been considered?

9.Has there been any consideration of how this model could be deployed in practical educational settings? Could it be integrated into a learning management system (LMS) or used by teachers for early intervention?

10.It is also suggested to add articles entitled “Aziz & Elsonbaty. Comparative Study of Different Classification Methods and Winner Takes All Approach” and “Blegur et al. Peer-Assessment Academic Integrity Scale (PAAIS-24)” to the literature review.

11.Did the authors compare their approach with standard educational prediction benchmarks (e.g., logistic regression, XGBoost, or LSTM)? This would help validate the unique value of the GDO-RBFN approach.

12.The conclusion mentions that the RBFN performed best with “educational habit features.” What specifically do these include? Are they related to study frequency, attendance, time-on-task, or something else?

13.The future direction toward integrating more advanced deep learning and exploring feature engineering techniques is well-aligned with current research trends. Could the authors elaborate on which specific models or techniques they plan to explore (e.g., transformers, attention-based networks)?

Reviewer #2: The paper is well written. The idea is interesting and the obtained result is valuable. This paper may be accepted if the following problems can be clarified.

1. What specific improvements the authors consider regarding the methodology?

2. Kindly enrich the literature review of this paper by citing additional references related to the topic addressed: Quantized Iterative Learning Control of Communication Constrained System with the Encoding and Decoding Mechanism, Transactions of the Institute of Measurement and Control; ADP-Based Prescribed-Time Control for Nonlinear Time-Varying Delay Systems With Uncertain Parameters, IEEE Transactions on Automation Science and Engineering; Saturated-threshold event-triggered adaptive global prescribed performance control for nonlinear Markov jumping systems and application to a chemical reactor model, Expert Systems with Applications; It could be the object of a brief consideration focused on the advances on the topic and make relation with this paper, which have to be discussed to indicate the contribution in the Introduction section, and in that way point out other contemporary approaches and possibilities.

3. Authors should argue their choice of the performance evaluation indicators.

4. Have the authors experimented with other sets of values? What are the sensitivities of these parameters on the results?

5. Explain the feasibility of the results from the implementation and computational point of view.

6. Authors should clearly define what quantities must be specified to begin the algorithm. Did the authors consider how many initial conditions affect the outcome?

7. The paper does not explicitly discuss the limitations of the proposed algorithm. Every algorithm has its shortcomings, and it would be helpful to include a section discussing potential drawbacks or scenarios where considered approach might not perform optimally.

8. Each reference has to be double verified, and also reference writing style needs to be uniform. It is essential to review all references, fill in any gaps with Volumes, Issues, and Pages, and revise any inaccurate information. Further, for current references that aren't yet listed in Volume and Issue, DOI numbers must also be added.

6. PLOS authors have the option to publish the peer review history of their article (what does this mean?). If published, this will include your full peer review and any attached files.

Reviewer #1: No

Reviewer #2: No

---

## [Author Response · Author response to Decision Letter 1]

5 May 2025

Responses to Reviewer Comments

Manuscript Number: PONE-D-25-15254

Title: Research on Learning Achievement Classification Based on Machine Learning

Authors: Jianwei Dong, Ruishuang Sun, Zhipeng Yan, Meilun Shi and Xinyu Bi

Dear Editor and Reviewers:

On behalf of all the authors, we express our deep gratitude to you for providing us with the opportunity to revise the manuscript. We acknowledge your valuable and constructive feedback on our work, ‘ Research on Learning Achievement Classification Based on Machine Learning’ (ID: IPM-D-24-00258). We have carefully reviewed the reviewers’ comments and have made every effort to revise the manuscript as suggested. The revised parts are highlighted in red throughout the revised manuscript. Herein we provide a detailed point-to-point response to the reviewers’ comments and concerns.

Responses to Reviewer #1

We sincerely thank Reviewer #1 for the detailed comments and valuable suggestions about our work. We believe these suggestions must be (and they have been) helpful for us to improve our work. Below we have collated the detailed and valuable comments by reviewer #1, giving our response and the changes we have made to the paper.

Q1: The study tackles an important and socially impactful problem—predicting student academic performance using machine learning, which has growing significance in the era of data-driven education.

Author Response: We sincerely appreciate the reviewer’s recognition of the significance and societal impact of our research. We agree that predicting students’ academic performance using machine learning techniques is of great importance, especially in the era of data-driven education. Your recognition will inspire us to further improve our work.

Q2: The proposed Gaussian Distribution-based Data Augmentation (GDO) technique coupled with a Radial Basis Function Network (RBFN) is interesting. However, is GDO a novel method or an adaptation of existing Gaussian-based oversampling techniques like SMOTE variants?

Author Response: We thank the reviewers for their interest in our proposed GDO-RBFN model. The GDO technique we use does draw on the idea of oversampling based on Gaussian distributions, but it is not a simple adaptation of existing Gaussian-based oversampling techniques (e.g., SMOTE variants). Specifically, GDO extends the original dataset by utilising a Gaussian distribution, and in doing so aims to address the problem of limited educational data and category imbalance. Unlike SMOTE and its variants, GDO generates samples based on the statistical characteristics of the data, not only focusing on a small number of class samples, but also emphasising on improving the model’s ability to deal with data with complex features while maintaining the original data distribution properties, especially in terms of non-linear relationships and multi-dimensional features, which makes the approach particularly suitable for the case of educational datasets with a limited number of sample sizes. Therefore, it is the GDO-RBFN that significantly improves the prediction performance of the student performance classification task.

Q3: The reported accuracy (94.12%) and F1 score (94.46%) are impressive. How does this compare with other baseline models without GDO? Were these improvements statistically significant?

Author Response: We thank the reviewers for recognising the accuracy (94.12%) and F1- score (94.46%) achieved in our study. To assess the effectiveness of GDO, we discussed two experimental settings separately, i.e., with or without GDO amplification. For example, Tables 4-7 show the results of comparisons at multiple baselines without using GDO amplification in different characterisations, and similarly, Tables 8-11 show the results at multiple baselines of comparisons after using GDO amplification. For example, Table 5 compares experimentally with Table 9. The chart numbers are all consistent with the manuscript.

Table 5. Classify by Personal Issues. The best results are highlighted in red and the second-best results are highlighted in yellow. (%)

Model Higher Education Students Performance Evaluation Dataset Student Performance

Accuracy Precision Recall F1-scores Accuracy Precision Recall F1-scores

ML LR 80.00 68.34 80.00 73.64 81.52 83.25 81.52 85.29

SVM 73.14 68.28 73.14 70.56 78.37 72.06 78.37 74.30

RF 80.00 72.11 80.00 75.51 83.75 75.23 83.75 77.35

DL MLP 77.29 68.77 77.29 72.63 79.59 70.76 79.59 75.93

ANN 77.29 68.70 77.29 72.51 80.12 84.08 80.12 81.54

BILSTM 77.24 67.93 77.24 72.19 82.94 85.66 82.94 84.78

RBFN 78.00 68.02 78.00 72.61 92.78 92.84 92.78 92.58

Table 9. Categorize by Personal Questions after Amplification. The best results are highlighted in red and the second-best results are highlighted in yellow. (%)

Model Higher Education Students Performance Evaluation Dataset Student Performance

Accuracy Precision Recall F1-scores Accuracy Precision Recall F1-scores

ML LR 77.23 77.01 77.23 76.10 77.09 76.74 72.09 78.09

SVM 87.50 88.76 87.50 86.93 85.69 86.93 85.69 84.71

RF 91.11 91.76 91.11 91.12 88.16 89.34 88.16 90.42

DL MLP 88.33 89.37 88.33 87.84 89.52 88.05 89.52 90.10

ANN 87.78 88.75 87.78 87.29 85.06 86.40 85.06 86.26

BILSTM 87.79 88.80 87.79 82.27 86.09 89.71 86.09 89.04

RBFN 90.83 91.97 90.83 90.37 91.25 90.27 91.25 92.03

Regarding statistical significance, the reliability of the synthetic data was verified through variance homogeneity and p-value, and we found that the variance homogeneity and p-value of the characteristics of the overall dataset, personal problem, family issues, and educational habits were higher than 0.05, especially the p-value of the overall dataset, which was 0.8387, suggesting that these improvements were statistically significant and meaningful.

Q4: It is suggested to add article entitled “Al-Ali et al. Analyzing Socio-Academic Factors and Predictive Modeling of Student Performance Using Machine Learning Techniques” to the literature review.

Author Response: Thank you for your valuable suggestions and we recognise “Analyzing Socio-Academic Factors and Predictive Modeling of Student Performance Using Machine Learning Techniques” article as an important contribution to this research area.

Author Action: Based on your suggestion, we have added this section to the literature review and cited it in the body of the manuscript, with the order of the literature remaining the same as in the manuscript. The manuscript is relevantly described as: The more typical K-means clustering and Long Short-Term Memory (LSTM) networks are used to model and predict student performance based on their reported behaviours and preferences [14].

[14] Al-Ali R, Alhumaid K, Khalifa M, Salloum S, Shishakly R, Almaiah M. Analyzing Socio-Academic Factors and Predictive Modeling of Student Performance Using Machine Learning Techniques. Emerging Science Journal. 2024;8:1304–1319. doi:10.28991/ESJ-2024-08-04-05.

Q5: While the classification accuracy is emphasized, does the study address interpretability of the models? Which features were most influential in predicting academic achievement, and can these be translated into actionable insights for educators?

Author Response: We thank the reviewers for raising the key issue of model explainability. In this study, RBFN was adopted as the core model, student performance is affected by the interaction of multiple factors, compared with the performance of linear models that are difficult to handle, RBFN the model maps input features to a high-dimensional space and is able to capture the complex non-linear relationships between features. Moreover, the RBFN radial basis function and GDO are both based on Gaussian distribution, and the generated new samples can be seamlessly integrated into the local response mechanism of RBFN. As a result, the accuracy of the augmented RBFN is significantly better than other models. In addition, we talk about the classification effects of different feature combinations (overall features, personal problems, family problems, and educational habits) separately by comparatively exploring different feature combination experiments, and we have uncovered the importance of certain features through in-depth exploration of the experimental results. For example, through the data in Table 10, we can find that the most significant improvement in classification accuracy of educational habits features is 93.58% of RBFN accuracy after amplification, which also indicates that these features of educational habits feature group are crucial for achievement prediction. This allows educators to design targeted interventions based on identified characteristics such as study habits, family background, etc., providing educators with prerequisite experience.

Q6: The paper mentions using variance homogeneity and p-values to evaluate the synthetic data quality. Could more advanced metrics like distribution overlap, Fréchet distance, or t-SNE visualizations be considered for assessing augmentation effectiveness?

Author Response: Thank you for your valuable suggestions, we strongly agree with you and believe that the introduction of more advanced metrics is necessary.

Author Action: Following your suggestion, we have also included distribution overlap and Fréchette distance as metrics for assessing the quality of the synthetic data, and the results are summarised in Table 12. Similarly, additional analyses on distribution overlap and Fréchette distance have been conducted in the Discussion section of the manuscript to provide a more comprehensive analysis of the effects of data enhancement. The chart numbers are all consistent with the manuscript. The manuscript is relevantly described as: Since the data were synthesised using GDO, we conducted experiments in order to verify the credibility of the synthesised data, using the Homogeneity of Variance, P-Value, distribution overlap and Fréchette distance as evaluation indicators, and the results of the experiments are shown in Table 12. Starting from the Homogeneity of Variance and P-Value, the Homogeneity of Variance and P-Value of the four categories are higher than 0.05, with the P-Value of the overall dataset being 0.8387, personal problems 0.9210, family problems 0.9075, and educational habits 0.9125. This indicates that there is no significant difference in the homogeneity of variance of the data in different categories, and the hypothesis of homogeneity of variance is valid. significant difference and the assumption of variance chi-square is valid. In terms of distribution overlap and Frechette distance, the distribution overlap values of the four categories are distributed between 0.94 and 0.97, close to 1, indicating that the distributions of raw and synthetic data on each feature are highly overlapped. And the Frechette distance is distributed between 0.02-0.04, all less than 0.05, indicating that the distribution of synthetic data and original data is very similar, suggesting that the synthetic data is of high quality. In summary, the distributions of the synthetic data on these categories were consistent with the original data with high confidence, further verifying the effectiveness of the GDO amplification method in generating credible synthetic data.

Table 12. Reliability Verification of Synthetic Data.

Homogeneity of Variance P-Value Distribution overlap Fréchet distance

Overall dataset 0.9177 0.8387 0.9512 0.03

Personal problem 0.9658 0.9210 0.9705 0.02

Family issues 0.9516 0.9075 0.9400 0.04

Educational habits 0.9529 0.9125 0.9617 0.03

Q7: Did the original dataset suffer from class imbalance (e.g., high vs. low-performing students)? How does GDO compare to traditional oversampling techniques like SMOTE, ADASYN, or GAN-based augmentations?

Author Response: In our study, the original dataset suffers from a class imbalance problem, where students’ academic performance, which we measure on a scale of 0-20, is related to course subjects. We study student performance as a separate feature during our experiments, so this imbalance problem does not affect the ability of GDO-RBFN to identify a small number of class samples. GDO has the following advantages over traditional oversampling techniques:

1)GDO generates samples by modelling the statistical characteristics of the data, which effectively reduces noise and improves the quality of synthetic samples.

2)Unlike methods such as SMOTE, GDO can effectively avoid generating overly dense data points, making the generated data more consistent with the actual distribution.

3) GDO can better handle complex multidimensional features, especially in terms of nonlinear relationships and high-dimensional features.

We verified the effectiveness of GDO in generating plausible synthetic data in our experiments by advanced metrics such as variance homogeneity, p-value, distribution overlap, and Frechette distance, as shown in Table 12. The chart numbers are all consistent with the manuscript.

Q8: The paper acknowledges the limitation in generalization to small datasets or specific student populations. Has cross-validation across multiple cohorts or schools been considered?

Author Response: We acknowledge that there are limitations in the ability to generalise to small datasets or specific student cohorts. In order to further enhance the generalisation ability of the model and to verify its applicability across different cohorts, in fact, a 10-fold cross-validation was conducted during the study, specifically, the original dataset was divided into 10 mutually exclusive subsets, and each time, 9 of them were used as the training set, and the remaining 1 subset as the test set. The process was repeated 10 times to ensure that each sample had the opportunity to participate in the evaluation as a test sample. With this strategy, the problem of limited generalisation ability due to small samples is mitigated to some extent, and the stability and reliability of the model under different data distributions is enhanced.

Author Action: We have added a description of the 10-fold cross-validation to the Results section of the manuscript. The manuscript is relevantly described as: In order to further enhance the generalisation ability of the model and validate its applicability across different cohorts, a 10-fold cross-validation was carried out during the course of the study, specifically, the original dataset was partitioned into 10 mutually exclusive subsets, and each time, 9 of them were used as the training set, and the remaining 1 subset as the test set. The process was repeated 10 times to ensure that each sample had the opportunity to participate in the evaluation as a test sample. The experiments were based on pytorch to build the RBFN model and Adam was used for learning optimisation.

Q9: Has there been any consideration of how this model could be deployed in practical educational settings? Could it be integrated into a learning management system (LMS) or used by teachers for early intervention?

Author Response: Various feasible options have been considered on how to deploy the model in real educational settings, but there are certain implementation difficulties, which will be key to the next part of our research. However, we believe that the proposed GDO-RBFN model can be used by teachers for early intervention. By generating personalised reports on students‘ academic performance on a regular basis, teachers can identify students’ learning difficulties and take targeted measures in a timely manner. For example, based on the list of high-risk students predicted by the model, teachers can intervene early to provide additional counselling or resource support.

Q10: It is also suggested to add articles entitled “Aziz & Elsonbaty. Comparative Study of Different Classification Methods and Winner Takes All Approach” and “Blegur et al. Peer-Assessment Academic Integrity Scale (PAAIS-24)” to the literature review.

Author Response: Again, thank you for your valuable suggestions, both of which you have suggested, ‘Comparative Study of Different Classification Methods and Winner Takes All Approach’ and ‘Peer-Assessment Academic Integrity Scale’, are the most recent articles in this field of study and are important references.

Author Action: Following your suggestion, we have added this section to the

---

## [Decision Letter · Decision Letter 1]

19 May 2025

Research on Learning Achievement Classification Based on Machine Learning

PONE-D-25-15254R1

Dear Dr. Ruishuang,

We’re pleased to inform you that your manuscript has been judged scientifically suitable for publication and will be formally accepted for publication once it meets all outstanding technical requirements.

Kind regards,

Dr. Rajesh Kumar

Academic Editor

PLOS ONE

Additional Editor Comments (optional):

Reviewers' comments:

Reviewer's Responses to Questions

**Comments to the Author**

1. If the authors have adequately addressed your comments raised in a previous round of review and you feel that this manuscript is now acceptable for publication, you may indicate that here to bypass the “Comments to the Author” section, enter your conflict of interest statement in the “Confidential to Editor” section, and submit your "Accept" recommendation.

Reviewer #1: All comments have been addressed

Reviewer #2: (No Response)

2. Is the manuscript technically sound, and do the data support the conclusions?

Reviewer #1: Yes

Reviewer #2: (No Response)

3. Has the statistical analysis been performed appropriately and rigorously? 

Reviewer #1: Yes

Reviewer #2: (No Response)

4. Have the authors made all data underlying the findings in their manuscript fully available?

Reviewer #1: Yes

Reviewer #2: (No Response)

5. Is the manuscript presented in an intelligible fashion and written in standard English?

Reviewer #1: Yes

Reviewer #2: (No Response)

6. Review Comments to the Author

Reviewer #1: The Revisions are satisfactory in my opinion, and I would certainly recommend the Editors to Publish the Paper in their esteemed Journal.

Reviewer #2: The topic appears interesting theoretically, and the applications are academic. There were some issues that have now been resolved. The results obtained are promising and accurate. The paper deserves to be published in this form.

7. PLOS authors have the option to publish the peer review history of their article (what does this mean?). If published, this will include your full peer review and any attached files.

Reviewer #1: No

Reviewer #2: No

---

## [Editor Report · Acceptance letter]

PONE-D-25-15254R1

PLOS ONE

Dear Dr. Sun,

I'm pleased to inform you that your manuscript has been deemed suitable for publication in PLOS ONE. Congratulations! Your manuscript is now being handed over to our production team.

Kind regards,

on behalf of

Dr. Rajesh Kumar

Academic Editor

PLOS ONE